# ROYAL SOCIETY
# OPEN SCIENCE

evolution/computational biology/complexity

replicator, parasite, RNA world,
automata chemistry, artificial life

**Author for correspondence:**
Simon J. Hickinbotham
e-mail: simon.hickinbotham@york.ac.uk

# Nothing in evolution makes sense except in the light of parasitism: evolution of complex replication strategies

Simon J. Hickinbotham[1], Susan Stepney[2] and
Paulien Hogeweg[3]

[1]Department of Electronic Engineering, and [2]Department of Computer Science,
University of York, York, UK
[3]Theoretical Biology and Bioinformatics Group, Utrecht University, Utrecht, The Netherlands

 SJH, 0000-0003-0880-4460; SS, 0000-0003-3146-5401;
PH, 0000-0003-3392-9839

Parasitism emerges readily in models and laboratory experiments of RNA world and would lead to extinction unless prevented by compartmentalization or spatial patterning. Modelling replication as an active computational process opens up many degrees of freedom that are exploited to meet environmental challenges, and to modify the evolutionary process itself. Here, we use automata chemistry models and spatial RNA-world models to study the emergence of parasitism and the complexity that evolves in response. The system is initialized with a hand-designed replicator that copies other replicators with a small chance of point mutation. Almost immediately, short parasites arise; these are copied more quickly, and so have an evolutionary advantage. The replicators also become shorter, and so are replicated faster; they evolve a mechanism to slow down replication, which reduces the difference of replication rate of replicators and parasites. They also evolve explicit mechanisms to discriminate copies of self from parasites; these mechanisms become increasingly complex. New parasite species continually arise from mutated replicators, rather than from evolving parasite lineages. Evolution itself evolves, e.g. by effectively increasing point mutation rates, and by generating novel emergent mutational operators. Thus, parasitism drives the evolution of complex replicators and complex ecosystems.

## 1. Background

As Dobzhansky said: 'Nothing in biology makes sense except in the light of evolution' [1]. However, this begs the question: how to make sense of evolution? Naively, the 'currency' of 'fitness' is replication rate. Many mathematical, *in silico* and *in vivo* models

of evolution adopt this currency explicitly as their fitness criterion, and maximization of replication rate is seen as the outcome of the evolutionary process in some evolutionary experiments [2,3]. However, the complexity that has evolved in the biosphere appears to falsify fast replication being the dominant evolutionary trend: elephants clearly replicate more slowly than do bacteria.

Here, we document the evolution of replicators in the Stringmol automata chemistry [4–6] to study how complexity can evolve at the expense of replication speed. Each replicator is a short computer program, consisting of a string of opcodes, that can replicate other strings that bind to it. These strings do not copy themselves (unlike the case in well-known automata chemistry models of evolution, like Tierra [7], Avida [8,9] and others [10–13]) but instead copy other strings, which may be copies of themselves, but may also be non-replicators. In this way, these strings resemble the RNA 'replicases' of the hypothesized RNA world of prebiotic evolution [14–16]. In such systems, the emergence of 'parasites', which are replicated but do not harbour replication potential, is deemed inevitable in real life [17,18], in models [19], and in experiments [20].

In contrast to most models of the RNA world (but like RNA replication itself), Stringmol replication is an active process, taking time as the program executes, copying the opcodes one by one. This should strongly disfavour the evolution of longer, more complex replicators, and strongly favour the evolution of fast replicating 'parasites', which, having lost the replication code, would tend to be (much) shorter.

Stringmol does not set an *a priori* fitness measure via some 'task': fitness is intrinsic and implicit in the ability of a string to increase its number by being copied by another replicator. Early experiments with Stringmol rapidly evolve to extinction, due to the evolution of faster replicating parasites.

Such evolution towards extinction by faster replicating parasites is to be expected, unless the replicators are embedded in (transient) compartments [21,22] or in space, where spatial pattern formation, and thereby higher order selection, prevents extinction [15,16,23]. Here, we embed Stringmol in space. We study how an explicit replication mechanism (via the Stringmol reaction mechanism) evolves to cope with emerging faster replicating parasites when spatial pattern formation prevents extinction.

In the early stages of evolution, extinction still occurs in about half the cases due to almost-immediately emerging fast replicating parasites. However, where the system survives long enough for the well-known replicator–parasite wave pattern to emerge, fascinating long-term evolution unfolds. Intricate replication mechanisms evolve to cope with the parasites, including *slower* replication, higher mutation rates, and a self/non-self discrimination check. Diverse ecosystems evolve, with relatively long replicators, subdued parasites, and large population densities. The replicators may even subdue parasites to such an extent that the parasites become rare, and then the replicators, not needing the defence against parasites any more, lose their complex countermeasures and become simpler and shorter; but new parasites emerge, reversing this trend again. These processes are detailed below.

Thus we see that parasites, while traditionally seen as a threat to evolving replicator systems [19], are instead the means by which complexity can evolve, provided that spatial pattern formation prevents global extinction (see also [24–27]).

# 2. Results

## 2.1. Stringmol overview

We describe in detail the Stringmol automata chemistry as used to perform our experiments in the Methods section below. Here, we provide a brief summary sufficient to understand the results as presented.

A Stringmol string is a variable length sequence of assembly language instructions (opcodes). The language is designed so that the program cannot crash, no matter what sequence of opcodes is executed. Stringmols bind probabilistically based on their sequence, and their encoded program executes over the bound strings: the bound strings 'react'.

The system is initially 'seeded' with hand-programmed replicators. A seed replicator binds to another string, copies it one character at a time onto the end of that other string, then cleaves off the copied string. The overall reaction is $R + C \rightarrow R + C + C$, where $R$ is the seed replicator, and $C$ is the copied string (initially also a seed replicator).

In order to allow evolutionary experiments, point mutation is included in the system. The copy opcode executes with a small probability of mutation, by copying a different character, or inserting or deleting a character. Thus the initial population of seed replicators gradually mutates into other forms as they are imperfectly copied. These mutations result in more convoluted execution pathways,

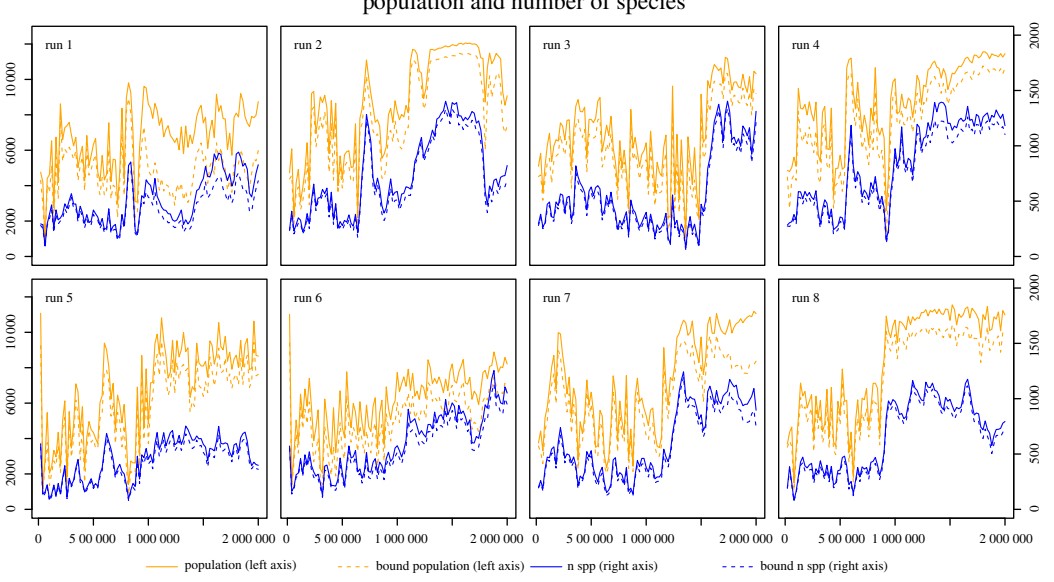

**Figure 1.** Population dynamics in the eight runs that survived to $T = 2\,000\,000$. Orange lines are population size (left axis) and show a general increase in overall reproductive efficiency as the environment is increasingly occupied; blue lines are number of species (right axis) and show a general increase in diversity.

including executing code on the other string; hence the result of a reaction can be a function of the program of both strings. We see forms of macromutation emerge through evolution.

We call a collection of strings with identical sequences a 'species'. All strings have a small probability of decay. Thus strings need to be replicated for the species to survive in the long term.

In previous work, we have studied Stringmol evolution in a 'well-mixed' system, where a string can potentially bind to and react with any other string in the system. Here, we conduct Stringmol experiments in a spatial arena: a two-dimensional grid of cells with periodic boundary conditions (a torus), where each cell either contains a single string, or is empty. Strings can bind only with other strings in their Moore neighbourhood. The cleaved product of a reaction is placed in an empty cell in that neighbourhood: if there are no empty cells available, the product string is discarded.

One kind of species that rapidly evolves into existence is the parasite. A parasitic pair of strings $R$, $P$ is one where $R$ can replicate $P$ but $P$ cannot replicate $R$. This results in a form of host-parasite dynamics, with parasites locally out-competing the replicators, and with both species evolving. Here, we study the evolution of these parasites, and the evolution of replicator defences against them.

## 2.2. Nothing makes sense …

We carried out 20 runs in total, of which 12 went extinct, and eight remained executing, after two million timesteps. Figure 1 shows the change in population size and number of species for the eight runs that survived to $T = 2\,000\,000$. The population generally increases in size over each run, due to a corresponding increase in replicating efficiency via evolution. The number of different species present also increases, usually in proportion to the total population size, but with some relatively sudden large changes in this ratio.

Increases in efficiency suggest identifiable changes in the mechanism of replication; there should be some observable features of the macro-level function of the strings themselves that correspond to these changes. One obvious explanation is that the programs become shorter in length or reaction time. The average values for these properties are shown in figure 2, and it is clear that this explanation does *not* hold. Although average program length decreases at first in all but one run, there is little or no corresponding decrease in reaction time; indeed, the latter half of all runs mostly show an increase in reaction time and program length. How has the system managed to increase its productivity without a corresponding decrease in reproduction time?

## 2.3. … except in the light of parasitism

Another possibility that could explain this increase in efficiency is that there is some secondary, emergent process serving to limit the population size increase early in the runs, and that this process becomes less

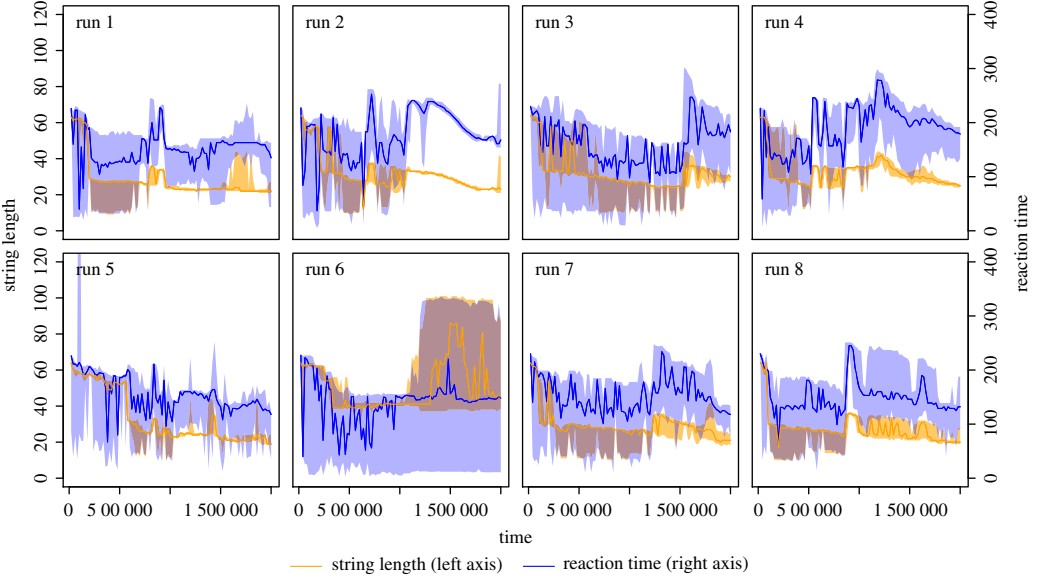

**Figure 2.** Change in program length and reaction execution time for molecular interactions. The midlines are the median length/time; the spread shows the interquartile range.

effective later in the runs. One such limiting process is well known: parasitism. Different entities in the system co-evolve different strategies to get themselves copied. Altruistic entities copy other strings, whereas parasites exploit their ability to bind to replicating programs and be copied, but do not return the favour. If parasites can be copied sufficiently faster than replicators, then the system will collapse as the number of available replicators diminishes, unless spatial patterns emerge that can protect a subpopulation of replicators from parasitism.

This proposed mechanism is difficult to discern from figures 1 and 2 for two reasons. Firstly, there is a significant role of spatial organization in the survival of the system that is not captured in these summary figures. Secondly, since parasitism is an emergent property of this system, we have to interrogate the system in order to identify the properties of parasitic entities.

We clarify the interaction between replicators and emergent parasites through detailed examination of a single run, before going on to identify the more general features and trends we have discovered via these experiments. Figure 3 shows the arrangement of emergent spatial patterns for run 2 of our experiments. The spatial arrangements of the population of strings are shown at six different time points, corresponding to different phases of the dynamics in the system. The percentage of parasitic reactions in the system drops to nearly zero by $t = 1\,500\,000$, and at this point the average program length is maximal. Note that at this point in the analysis, the function of individual molecules in the reactions is inferred by studying the emergent spatial organization of the system and by reference to dynamics observed in similar studies [15,16,23]. This inference has been confirmed by studying individual reactions as will be described below.

**t1**: shows the state of the system early in the run. At this point, there is a collection of replicator programs that follow the same basic execution pattern as the seed replicator. These are shown in red and orange. Parasites have already emerged at this stage and surround the trailing edge of these waves of replicators in the same manner described in e.g. [16,28].

**t2**: shows that the system has become organized into (pale blue) patches of short, mutually replicating strings of around the same length, with shorter (dark blue) parasitic entities around the margins. These replicators are more similar in length to the parasites, which confers an increased ability to compete for replicating resource. However, at this point, although the average program length has shortened, the reaction time has hardly changed (see top right panel of figure 3). How?

**t3**: shows the emergence of a longer replicator, as indicated by the greener patches in the figure, which demonstrates an uptick in both program length and reaction time compared with t2.

**t4**: although the difference in length is smaller, there still appears to be two distinct lengths of the replicator patches. Close inspection reveals two shades of green, and the yellower (longer) shade has fewer parasites.

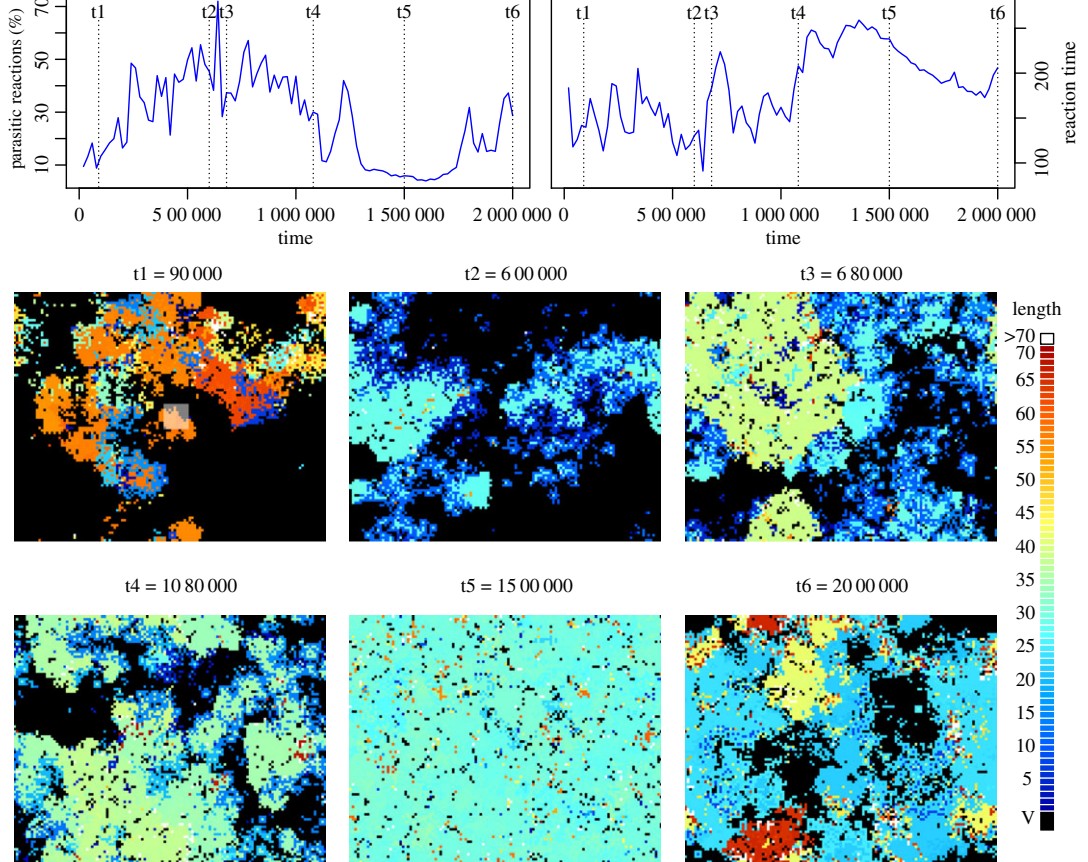

**Figure 3.** Change in program length and run-time for molecular interactions. Top panel shows change in the percentage of reactions which are parasitic (left) and mean reaction time (right). Bottom panel shows a visualization of the spatial arrangement of the strings at times t1–t6 during the run. The opaque square in the t1 image shows the initial distribution of the hand-coded replicator molecules. Each pixel in the image represents a cell in the grid. Empty cells (void, V) are shown in black. Occupied cells are coloured according to program length. Shorter strings are more blue, longer strings are more red. Where length is greater than 70 (in panel t6), cells are coloured white. The seed replicator is length 65.

**t5**: the system is at carrying capacity. The few empty cells are due to the random decay process in Stringmol. The string length has become shorter again, and the strings are arranged in patches of uniform length. Although these lengths are approximately the same as those for t2, it is clear that the system is more productive. This highly productive period is also a time when parasites have been driven nearly to extinction.

**t6**: at the end of this run, the number of strings diminishes concurrently with a re-emergence of parasitic strings and very long strings.

Thus we see that at each stage after t1, new behaviours have emerged that change the dynamics of the system. In order to understand these behaviours, we need to investigate the program execution structure of the key reactions at each of the time points described above.

## 2.4. Innovative strategies in run 2

The spatial patterning shown in figure 3 illustrates the dynamics of the system, but does not explain the selective advantage that each replication strategy has over others. In order to discover this, it is necessary to inspect the reactions between entities at the program execution level, by following the execution of the dominant reactions at each time point.

In order to determine whether properties like parasitism hold, which require the determination of a counterfactual behaviour (equation 5.2), we have to perform further analysis of the strings observed at certain epochs. See §5.3 'Reaction types' below for a description of how this is achieved. We summarize the resulting behaviours here.

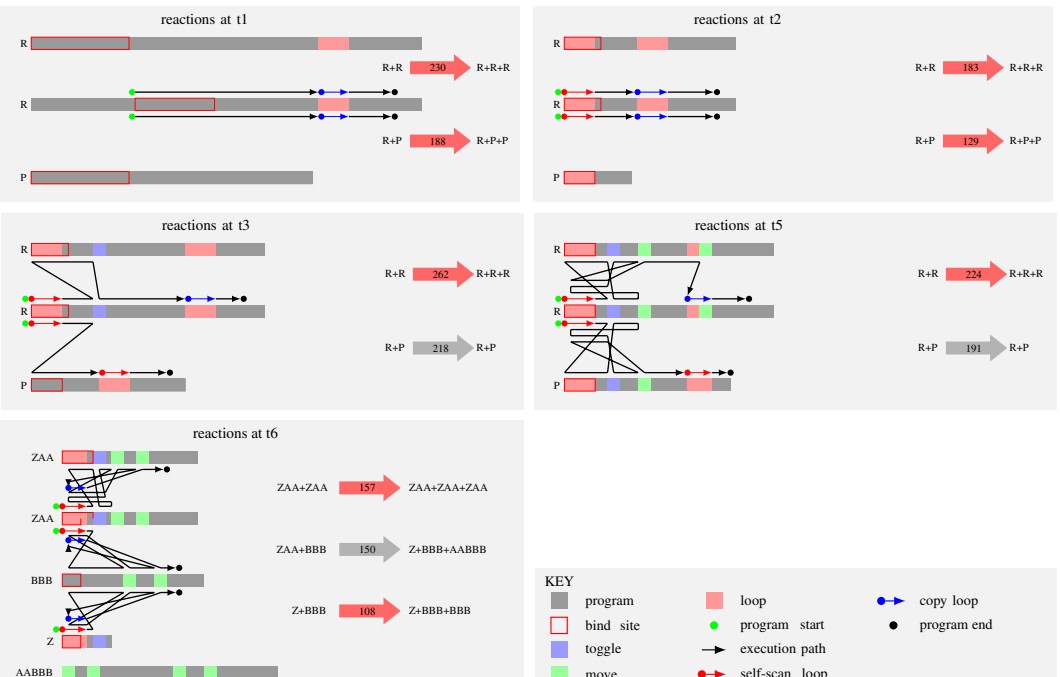

**Figure 4.** Evolved replicator mechanisms for t1, t2, t3, t5 and t6. The grey bars represent the program strings, with coloured rectangles highlighting various regions (see key panel). The first four panels illustrate how the replicator (centre string) copies two different classes of string: a replicator (top) and a parasite (bottom). The t6 panel summarizes several key reactions occurring at that time point. The length of the bind site is different for the 'ZAA+ZAA' and 'ZAA+BBB' reactions. The black line represents the program execution path (which opcode on which string is being executed at each time point). The fat arrows to the right summarize the reactants and products; the colour represents a replication (red) or other (grey) reaction in which neither of the input reactants are copied; the number shows the program timesteps.

Figure 4 shows program flow for a variety of different replication programs from the time points identified in figure 3. The t1 panel shows the self–self reaction for the original 'seed' replicator and with a parasite. The relative position of the complementary binding sites (red-edged rectangles) determine the entry point (start) of the program, shown as a green dot. Execution proceeds on the active string in a linear fashion (black line) until entering an iterative 'copy loop', shown as a pink box on the string and a red arrow with a red dot at the beginning on the program execution line. This loop is the heart of the replication process, producing a new string that is a copy of the template string. When copying is finished, the program proceeds to the exit instruction, indicated by a black dot, and terminates.

The t2 panel of figure 4 shows the dominant replicator at this time point. Selection pressure for shorter sequence length and hence faster replication is evident. Two further important adjustments to the replicating reaction program have emerged. First, the manner in which the strings bind is no longer complementary (because the bind uses opcodes which are specified to bind to each other), and there is a 50 : 50 chance of the program entry point being on either string. This has no effect on the efficiency of replication, but has important ramifications for parasitic strings that bind to the replicator. With complementary binding, it is possible for parasites to avoid binding to other parasites, since they need only one of the two complementary sequences to bind to the replicator. With the non-complementary binding here, parasites *do* bind to each other, which reduces their specificity for replicators, so giving replicators a selective advantage.[1] Second, an innovation we call *self-scan* has emerged, whereby execution enters a loop, but instead of immediately producing a new string, the program string initially overwrites itself with its own code, leaving its sequence unchanged (unless mutation on copy happens), before going on to copy the other string. The main effect is to *slow down* the entire replication process. The selective advantage of this feature is apparent only in the presence of parasites, since their rate of reproduction is *also* slowed. Shorter programs get copied more quickly because fewer iterations of the copy loop are required, and since parasites do not need to carry any functional code beyond the binding

---

[1]The analogy in RNA is that short repeats can effectively reproduce the effects of non-complementary binds at the macro level: for example, the sequence GCGCGCGCGCGC can bind to an instance of itself, albeit with a minor misalignment.

region, selection favours shorter parasites, which can destroy the entire system. The self-scan feature adds a fixed cost to replication which parasites *cannot* avoid, and it appears that (inefficient) replicators can persist in the presence of parasites by adopting this strategy. This explains how short programs can have long reaction times as observed in figure 2—a second iteration over the length of the program has been added. Finally, because self-scan is achieved by using the copy operator (which fires point mutations) to overwrite every opcode with the same opcode, the point mutation rate is approximately doubled for replicating reactions that have this feature.

The third panel of figure 4 shows the dominant reaction at t3, a time of sharp increase in population. The self-replicating reaction involves longer strings (and thus slower reactions) than those shown for t2. The program has evolved to move some execution to the partner string. The blue and green boxes on each string indicates a 'toggle' or 'move' operation, which switches execution to the partner. (Although these two operations have similar effects, they are derived from mutations of different parts of the original replicator code.) Where the string is bound to an instance of itself (a self–self reaction), execution is subsequently toggled back to the active string, and only then does execution enter the copy loop. Considered in terms of replication alone, this feature adds to the fixed overhead of execution initiated by self-scan. However, this toggling feature acts as a gatekeeper for replication: if a non-self string does not have the toggle code, execution is not passed back to the active string and replication does not happen. This effectively blocks parasitism by short strings: the parasites now have to do more than simply bind to a replicator in the correct place; they also need the toggle facility to return execution to the replicator and thus access the copy loop. The 'P' molecule in the t3 panel was able to parasitize earlier replicators, but it cannot do this with the dominant replicator at t3, so it is not copied. Furthermore, as the figure shows, the code of the parasite is executed; in this case, a secondary self-scan occurs. So parasites are not replicated, but they are held in an unproductive reaction for long periods of time during which they are unable to bind to other replicating strings that do not have the toggle protection.

The competitive advantage gained via this strategy is short-lived, because new parasite strings quickly evolve by mutation from these replicators to incorporate the toggle function and access the copying mechanism. From now on, there is an arms race (but not the classical evolutionary 'Red Queen' arms race [29], because the new parasites evolve from the replicators, rather than evolving in their own lineage), in which the replicators add increasingly sophisticated security schemes around the copy loop and parasites overcome them. In the run we are studying here, we see a point where the replicator wins this race, because at time t5 in figure 3, the population is highest and parasites are almost extinct. This is also evidenced by the narrow interquartile range for both string length and reaction time during the latter half of run 2 in figure 2, where replication is the unimodal behaviour. The t5 panel of figure 4 shows the dominant program execution at this point: here we have two additional checks on the partner string's code that must be passed by a parasite in order to access the copy loop. The effect is that parasites are shut out of replication if they do not have these checks, but even with these checks they have no advantage in terms of replicative efficiency because they are not much shorter than replicators. So we see a corresponding drop in the number of parasitic reactions: only 323 of a total of 5671 reactions at t5 are parasitic.

Once parasites become rare, the selective pressure to counter them is removed. Efficiency once more becomes the dominant mode of selection, until parasites emerge anew. This is the situation in the t6 panel of figure 4, where the interactions between dominant strings are shown. The dominant self-replicator string, dubbed ZAA to indicate its component substrings, is shorter than the strings at t5, but the self–self reaction takes longer. A single loop structure is responsible for both self-scan and replication in 157 timesteps. Although the parasite dubbed BBB has the code to return execution back to ZAA, it does not position the cleave point in the correct place for replication. As a result, the tail end of ZAA is appended to the front of BBB to form AABBB as the product of the reaction, leaving a shorter molecule Z, which cannot self-replicate (the Z:Z reaction is not shown in the figure). When Z binds to BBB, a replication reaction is possible, because BBB contains the program fragment needed to arrange the pointers to allow replication; thus BBB catalyses the creation of Z, and then parasitizes it. Inspection of the spatial arrangement of strings at time t6 in figure 3 shows short (dark blue, Z) strings surrounded by longer (pale blue, BBB) ones, reflecting the mechanism just described. If the short partner goes extinct, the strategy is still advantageous as long as strings that carry out self-scan can be bound to.

## 2.5. Reproductive efficiency

Figure 5 shows the potential increase to maximum population size given the rate of replication, arena size and decay rate for the seed replicator and the dominant replicators just after t1 and then for t2–t6. The t1

royalsocietypublishing.org/journal/rsos　　R. Soc. Open Sci. **8**: 210441

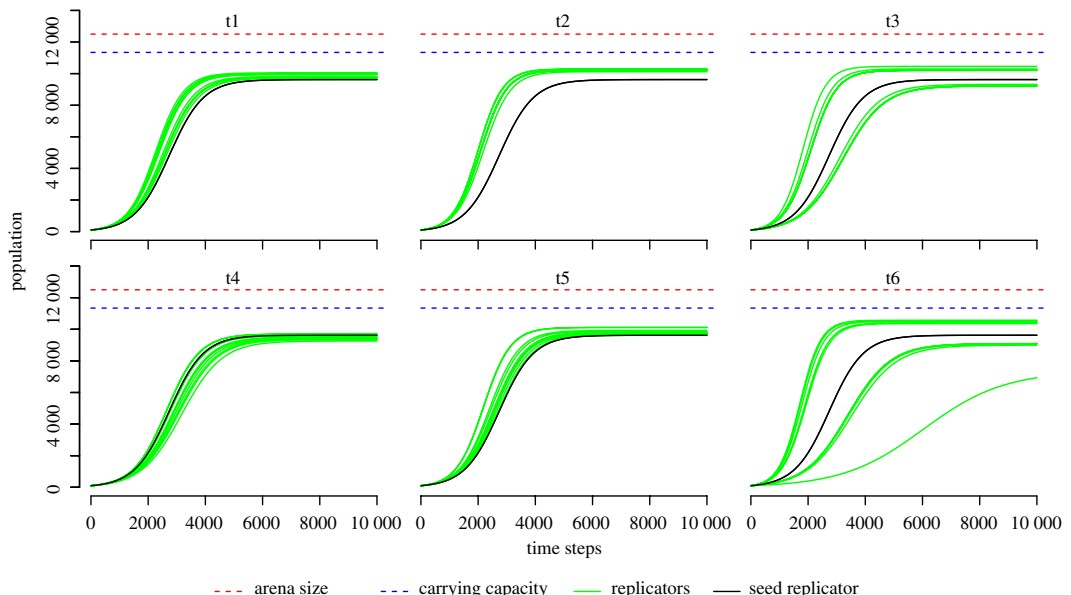

**Figure 5.** Change in reproductive efficiency and maximum possible population size for t1–t6. Each line shows the rise to maximum population size from a population of two self-replicators in a well-mixed system with no mutation. Each green line is for a common replicator species in its respective time point. The black line is for the seed replicator.

and t2 time points show increase in reproductive efficiency: the green lines (evolved replicators) rise more quickly (reproduce more efficiently) than does the black line (seed replicator). At t3, there are two groups of replicators: one group is more efficient than the seed and one has parasite protection, which comes at a reproductive cost. By t4, the slower strategy has come to dominate via this mechanism. Although all the t5 replicators increase more quickly and have a higher carrying capacity than t4, the difference is relatively small: it is the evolved resistance to parasitism that explains how the population size is larger than seen before. By t6, we see the return of the mix of fast replicators and mechanisms to combat parasitism as in t3, only here the difference between the two strategies is more pronounced.

## 2.6. Macromutations

Changes in the length of the replicators play an important role in the dynamics of the system. Stringmol defines only point mutations, but here large changes are happening, by an emergent mechanism different from the explicitly coded point mutations.

How do these large changes between these strings arise as they evolve? A series of single-point mutations appears to be a highly improbably route to the observed rearrangements of the program instructions that yield the innovative behaviours. By studying the sequence of products from long lineages of replicating reactions, we have found that these changes are consequences of single point mutations combined with subsequent changes in binding and cleaving of strings at points different from the original design.

Some single point mutations change the function of the program such that a cascade of mutant species are subsequently produced, none of which is self-sustaining, but which eventually result in new replicators that are able to increase in number. As an example of these mutational cascades, figure 6 illustrates stages in the transition to the first self-scanning replicator.

1: The seed replicator, divided into four regions: two complementary bind sites in yellow and grey, the copy loop in red, and the cleave/end region in blue.
2: A mutation in the positioning of the write pointer causes the trailing three regions of the seed replicator to be copied onto another replicator string. In the resulting outputs of the reaction, the first bind site is lost, and the copy and terminate regions are repeated. This string persists for some time, and fixes by copying itself onto whatever it is bound to.
3: A mutation in the positioning of the program flow pointer for the cleave operation means that the string is further truncated.

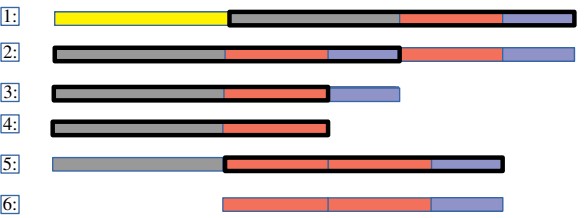

**Figure 6.** Macromutations to new behaviours: stages in the transition from the hand-designed seed replicator to the first self-scanning replicator. The black box outlines the part of the string that is written to the new mutant via the copy loop in the replication program. The colours indicate particular substrings (see text for details).

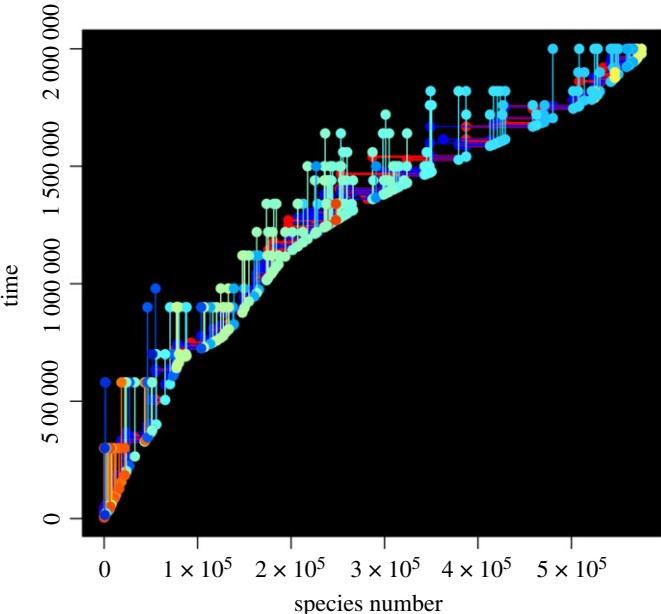

**Figure 7.** Phylogeny of dominant strings at end of run 2. The *x*-axis is species number, indexed by order of appearance in the run. Only strings involved in generating new species in the ancestry of these replicators are shown. Vertical lines indicate the first and last appearance of each species in this ancestry, coloured by species length as in figure 3. Horizontal lines indicate the 'parent' species of each new species; red indicates the string with the program entry point for each reaction, and blue indicates its partner.

4: A second repositioning of cleave truncates the string again.
5: The truncated string from step 4 is copied onto the start of a longer string that also has the red and blue regions at the end of its sequence.
6: Finally, a further mispositioned cleave truncates the string from 5, but here the trailing portion of the string forms a viable replicator, and the copy loop additionally functions as the bind site.

## 2.7. Replicator phylogenies

Tracing the ancestry of strings in an RNA-world scenario is challenging: the definition of species is not possible; although Stringmol has the potential to form clades that cannot replicate each other, this has not been observed. A member of a species (a specific sequence of opcodes) may be the product of several different reactions; as long is it can *bind* to a replicator (and overcome any mechanisms to prevent it), it has a chance of being reproduced. In this context, we attribute the creation of a new species, and the strings that generated it, to the reaction that caused its first appearance. A graphical depiction of the 'phylogeny' of the ten most populous replicating strings at t6 for run 2 is shown in figure 7. The lower edge of the species in these plots indicate the rate of production of new species: a relatively steep angle indicates a relatively slow production of new species. As the self-scan feature is introduced, the rate of production increases: self-scan has the possibility of inducing mutations while scanning. The length of the strings also has phases, from the relatively long seed, through a short phase of efficient copying, and then with the introduction of mechanisms to protect from parasitism.

This shows that new parasites emerge as offspring of replicators either directly or as the result of short mutational cascades—separate lineages of parasitic molecules are not a feature of this system.

## 2.8. Parasite phylogenies

A picture of the phylogenies of parasites would not save a thousand words: overwhelmingly, parasites are direct mutations from interactions between replicators. Any separate lineages of parasites that we have observed are short-lived cascades from a viable replicator population to a viable parasite that fixes via non-viable intermediates.

A novel parasite may flourish because it inherits features of its replicator ancestor, by which it may circumvent that replicator's defence system, in particular its self–non-self check. When the replicator evolves an effective defence against these parasites, they are replaced by newly parasitic offspring of the replicators, rather than by their own mutated offspring. The emergence of novel parasites from replicators are 'easy' mutations in our system, as it would be in the RNA world as well. Thus the dynamics observed here of continuously emerging novel parasitic lineages should probably be the case in the RNA world as well, rather than a classical arms race between co-evolving independent replicator and parasitic lineages that have been postulated and observed before. The active replication process implemented in our model can 'invent' the appropriate mutation, whereas in most models of the RNA world such mutations are not possible. Recent *in vitro* RNA evolution experiments [30] have observed similar evolutionary dynamics to those observed here.

## 2.9. Common trends and emergent mechanisms

Exhaustive analysis of all runs shows a striking common strategy to accommodate parasitism: deceleration of reproductive rate combats parasitism by diminishing the rate advantage that parasites gain when they jettison their reproductive code. There are many ways that this strategy can be achieved, and this has led to a diverse behavioural oeuvre in these experiments.

Features common to all runs are: a move to *non-complementary binding*, which has the effect of forcing 'hostile' sequences to bind to each other, thus reducing their access to replicators; *self-scanning* as a way of increasing the reaction time in a way that is not proportional to the length of the string being replicated, reducing the benefit of evolving to be short; diverting the *execution path* such that repeated excursions to the partner string demand that it reciprocates, thus demanding that replicated molecules adhere to particular structural requirements.

In addition to features seen in all runs, there are some runs dominated by short *hypercycles*, where *A* copies *B*, *B* copies *A*, but neither *A* nor *B* can self-copy: *A* and *B* are mutual replicators but not auto-replicators.

# 3. Discussion

## 3.1. The counterintuitive rise of complexity

In the absence of other selection pressures, it might be assumed that self-replicating entities would evolve to be replicated as rapidly as possible, in order to gain advantage over competing entities that are replicated more slowly. This drive to efficiency might be thought to dominate fitness, leaving no room for an increase in the complexity necessary for the emergence of novel behaviours. However, as we see, both in the real world and in *in silico* experiments, complexity, and size, does increase. The causes are subtle, and need a specific style of *in silico* experiment in order to unpick the details.

## 3.2. Stringmol compared with other model systems

Experimental vehicles to explore replicator-parasite systems are complex to set up. *In vitro* experiments have problems such as: designing the seed replicator; maintaining the correct concentration of resource materials throughout the experiment; interpreting the evolving concentrations of replicator species and their cohorts. Although *in silico* experiments allow all environmental variables to be controlled with precision, and the results to be fully interrogated, the challenge is that a complete simulation environment, including the population of replicating entities, must be constructed *ab initio*, and detailed experiments can be computationally intensive. Work to date on *in silico* systems can be categorized into two approaches, which we call 'explicit' and 'emergent'.

The explicit approach focuses on interactions where the rate of replication is not subject to evolution; the evolutionary potential of replicators and parasites is similar. In some such models [16,24,28], only the parameters determining the interaction strength of predefined or emerging lineages evolve; in RNA-sequence based models [31,32] the interactions evolve through its primary and secondary structure. Because the replication is abstracted, these experiments can be large in scale, allowing many tens of thousands of individuals to be modelled in detail over long timescales. These experiments demonstrate emergent macroscopic spatial arrangements of replicators and parasites preserving replicating populations that would be driven to extinction in well-mixed systems.

The emergent approach [7–9,33–35] focuses on systems where the rate of the replication emerges from a detailed implementation of a replication mechanism. The computational costs mean that these experiments are relatively small in scale; however, the complex dynamics of the replication process itself can be studied. There are relatively few emergent systems, and mechanisms by which parasitism emerges is different from in Stringmol, for two main reasons. Firstly, the mechanism for reproduction is different: program instances in the other systems reproduce themselves, by self-inspection (only one string is involved), whereas Stringmol reproduces a separate instance (two strings are involved). Also, the other systems' access to computational resources violates locality in space and the definition of 'self': individuals can access the resources (CPU cycles and memory space) allocated to others, forming the definition of parasitism in those systems.

In both the explicit and the emergent approaches, the effects of parasitism on the dynamics of the system have been reported. However, each approach has limitations. In the large-scale simulations with abstracted replication, since replication is not modelled directly, it is not possible to investigate the effect of mutation on the replication process itself. In simulations where replication is explicitly modelled, the systems tend to impose explicit fitness functions to either maintain the replicating population [7] or to explore the evolution of functional properties built on top of the replicating component [27].

The work reported here, Stringmol with spatial position, combines the advantages of each approach. Previous experiments with aspatial Stringmol also exhibit parasites and other complex behaviours [6], but because the system is well-mixed, parasites can drive the system to extinction at any time. As in earlier spatial models of the RNA world, here extinction due to the inevitably arising parasites is prevented by spatial pattern formation of chaotic wavefronts where replicators invade empty space and are out-competed by the parasites at the back of the waves. The dynamics of the waves impose similar selection pressures in the various models. Subsequent evolution copes with these selection pressures in different ways in the different models, dependent on the evolving entities' degrees of freedom.

## 3.3. Selection pressures

Since the replication program itself evolves, selection can act directly on the replication strategies in the system. This allows the replicators to have a much greater evolutionary potential than the parasites, and they dominate the evolutionary process. This has a profound impact on the evolutionary dynamics.

In almost all previous models, a long-term co-evolutionary process of two or more independent lineages unfolds. There, new parasites originating from mutated replicators cannot compete with the specialized parasitic lineages. By contrast, in the model here, parasites with novel functionality do not arise from mutated parasites. Instead, throughout the evolutionary run, mutation causes replicators to lose their replication capacity by having part of their program deleted; they mutate into novel parasites that inherit features of the replicator, and therefore may 'fool' the self–non-self discrimination defences of the replicators. In recent RNA evolution experiments [30] both these types of parasite evolution are observed: a parasite lineage emerges very early, persists and evolves continuously; also new, shorter-lived parasitic lineages arise from mutated replicator strands and exploit the mechanisms those have evolved to increase their own replication.

Similar selection pressures in all these models force the evolutionary processes, but can be achieved by different means, or lead to different outcomes. For example, in [28], the *parasite* evolves a parameter that decreases its relative competitive advantage, whereas in the model here, a weaker relative advantage of the parasite is a result of evolution of the *replicator*: its self-scanning mechanism slows down replication. This mechanism can occur only in a system where replication can take different lengths of time. This self-scan is apparently squandering resources in order to avoid parasitism: there is selective advantage in actively wasting resources outside of the main replicating activity. By evolving a process in addition to that of pure replication, the resources that are 'squandered' provide an evolutionary toehold for directly advantageous processes to develop, giving a route to escape the relentless pressure to simply replicate ever more efficiently.

Another important selection pressure is avoiding the formation of complexes that prevent replication, for example parasite–parasite complexes that hinder parasite replication. In the model here, the replicators evolve non-complementary binding, which ensures that parasites cannot avoid forming such complexes. In the sequence-based RNA models [32], with complementary binding, the parasites evolve a folding structure that prevents such complexes, and parasites are always available for replication.

In the model here, the replicators evolve such intricate recognition programs that parasites are (almost) extinguished. However, without the selection pressure of the parasites, these elaborate mechanisms are then lost, and new parasites emerge. In the sequence-based RNA-world models, parasite lineages cannot be sustained at very high mutation rates [31,32], and the replicators evolve in such a way that no parasites can be formed by single, or even multiple, mutations.

These parallels and contrasts underline the need for studying evolutionary process not only in terms of preset selection pressures, but also by exploring how the structural background can shape the outcome.

# 4. Conclusion

We have demonstrated a pathway for how and why a replicator might involve itself in complex activities other than pure efficient copying. First, it is advantageous to the individual replicator to 'cheat' and become a parasite. Next, the remaining replicators slow down their replication rate and introduce fixed costs that replicators and parasites alike must pay. Then the resources used in paying the fixed costs are themselves subject to adaptation. It is not implausible that such evolving emergent mechanisms be exapted into other functionality too.

Evolution in replicator systems makes sense if we consider the selection pressures induced by the emergence of parasitism.

# 5. Methods

## 5.1. Stringmol details

Stringmol [5,6] is an automata chemistry [36] in which the 'molecules' are programs encoded as strings of *opcodes*: sequences of symbols, each of which specifies a computational operation to be performed. The sequence composition determines the bind probability, execution pathway and product(s) of the reaction. Details of the Stringmol language and execution semantics are given in [5]. Here, we describe the main features.

**Opcodes**. The Stringmol assembly language has 33 opcodes. Seven of these, ?$^%}>=, are functional: they manipulate pointers, copy symbols, loop and cleave strings. The remaining 26 opcodes, the characters A–Z, have no operation when executed (they are 'no-ops'), but are used when binding strings, and as modifiers of the functional opcodes.

**Pointers**. Each bound string is supplied with four pointers (instruction, flow, read, write) for it to use as it executes. These provide the only program 'memory' in addition to the strings themselves: there are no registers or stacks to hold values.

**Binding** to form reacting pairs is probabilistic, based on the strength of string matching determined by a Smith-Waterman algorithm. The no-ops have complementary matching based on ROT13: A matches N, B matches O, and so on. The functional opcodes have non-complementary matching: each functional opcode matches itself.

When two strings bind, one is designated as string1 and the other as string2. This designation depends on the lengths of the substrings before the bind site. If the bind is 'asymmetric' (the bind site is nearer the start of one string than of the other), then the string with the longer end before the bind site is string1. If the bind is 'symmetric' (the bind site is the same distance from the start of both strings), then string1 is chosen randomly. This leads to three possibilities when attempting to bind two strings A and B: (i) no bind, A and B cannot react; (ii) binding site such that A is always string1 and B is always string2; and (iii) binding site such that either A or B can be string1.

**Mutation** happens stochastically with a preset probability when a program executes the copy opcode '='. On mutation, the symbol at the read pointer is miscopied: a randomly chosen different symbol is written at the location of the write pointer. (See [37] for an investigation of behaviours when the miscopying is biased rather than random.)

**Decay** also happens stochastically, removing strings from the system with a fixed uniform probability each timestep. This frees up space for new stringmols, and ensures species of strings must be actively

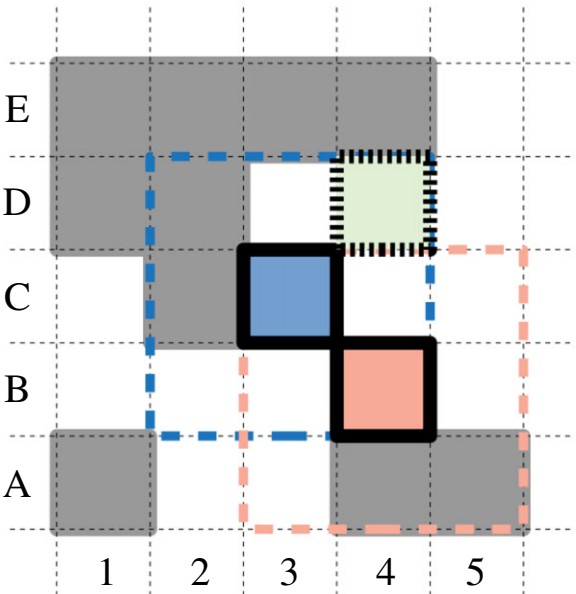

**Figure 8.** Reaction protocol. See text for details.

reproduced to maintain their presence in the arena. It is possible for the entire arena to 'die' if the community of strings is no longer self-maintaining.

**Reactions** between strings occur by executing the sequences of opcodes of strings in the reacting pair. On binding, four pointers that control program execution are initialized. The start point of the reaction program is the end of the bind site on string1. The program executes, with one opcode executed per timestep, using opcodes of one or both strings, depending on the sequences. (Previous work has an energy model: opcodes are executed only if energy is available. Here, all opcodes waiting are executed, with a limit of one execution per reacting pair per timestep.) For a given reaction, one opcode is executed each timestep, until either the reaction program terminates, or probabilistic decay occurs. Execution and effects are purely local to the pair of strings in the reaction, and any product strings that result.

**Arena**. This work introduces 'spatial Stringmol', in contrast to the 'well-mixed Stringmol' of earlier work. The strings are placed on a two-dimensional toroidal grid, and reactions are permitted only between individuals that are in the Moore neighbourhood of each other. New products of reactions are placed in empty cells around the first string in a reaction, or discarded if there are no free cells available. An example of a spatially constrained reaction is illustrated in figure 8. Empty cells are shown in white, occupied cells are shown in grey. **1:** cell C3 (blue, solid border) is selected randomly, and initiates a bind with a randomly chosen unbound string in its Moore neighbourhood (indicated by the dashed blue box). **2:** a bind is achieved with cell B4 (pink, solid border). String1 is chosen as the entry point for the reaction from the position of the bind site on each string. **3:** during the reaction, any new product strings are placed in an empty cell in the Moore neighbourhood of string1, shown as the blue hatched box. In this example, C3 is string1, so products can be placed at D3, D4, C4, B2 or B3. Had cell B4 been string1, then products could have been placed within *its* Moore neighbourhood, shown as the pink hatched box. **4:** in this example, a new string is produced, and placed in cell D4. This new string is available for binding on the subsequent timestep. **5:** The two original strings remain bound until the program terminates.

## 5.2. Configuration

In the experiments reported here, there are 20 runs with different random number seeds. The arena size is 12 500 cells, arranged on a toroidal grid, size 100 × 125 for 10 runs, and 250 × 50 for 10 runs. Each run is initialized with 100 'seed replicator' strings with sequence `WWGEWLHHHRLUEUWJJJRJXUUUDYGRHJLR` `WWRE$BLUBO^B>C$=?>$$BLUBO%}OYHOB`. Ten runs place these strings in a 10 × 10 block, and 10 runs place them in adjacent pairs at 50 random positions throughout the grid. This gives four treatments with five replicates each.

The runs were done on a Sun Grid Engine at the University of York. This facility has a CPU time limit of 5 days on individual runs, so it was necessary to restart runs until a simulation clock time of 2 million timesteps was reached; CPU time versus wall clock time for runs varies depending on how many

reactions are executing in the arena at each timestep. The log files for a run are a concatenation of the separate restarts runs. The runs were performed with Stringmol software v. 0.2.3.4, available at github.com/uoy-research/stringmol/releases/tag/0.2.3.4.

## 5.3. Reaction types

As the results show, these runs generate many emergent behaviours that were not predefined, including non-complementary binding, parasitism and various different patterns of replication. In order to automate the classification process, it is necessary to define these properties rigorously. See [38] for a full formal description; relevant properties are summarized here.

These properties are defined in a different context from the experimental runs, in that mutation is switched off during analysis. So a given reaction behaves deterministically during analysis, allowing its type to be defined, whereas it might exhibit different stochastic mutations during an experimental run. The reaction types used in the analysis above are:

**Replicator**. In a replication reaction, one of the strings is *replicated*: there are more instances of that string after the reaction than before. In the simplest case, of the initial seed replicator, we have a reaction like $R + T \rightarrow R + T + T$: $R$ is the replicator, and $T$ is replicated template. Later, more complicated replication reactions may be seen.

We say that a reaction has the *repl*($R$, $T$) property if $R$ replicates $T$. $T$ might be the string2, replicated by string1 $R$, or it might be string1, using the code of string2 $R$ to copy itself, or there might be some more complicated case using the code on both strings. We cannot tell where the replication code lies in either case and the definition does not require it to be known. Whether or not $R$ is string1, whether or not $R$ carries the copying code, it is the 'catalyst' for $T$'s replication. So, irrespective of which is string1, we say '$R$ replicates $T$', call $R$ the replicator, and $T$ the template.

**Auto-replicator**. In an auto-replication reaction, a string can replicate (another instance of) itself; there are at least three copies of the string after the reaction.

$$\text{auto-repl}(R) \triangleq \text{repl}(R, R). \tag{5.1}$$

The initial seed replicator is an auto-replicator.

**Parasite**. In order to determine that a string is a parasite, we need to consider the reaction between two strings 'both ways round', that is, what happens when each string is string1. The parasitic property holds if $R$ replicates $P$, but $P$ does not replicate $R$:

$$\text{parasitic}(P, R) \triangleq \text{repl}(R, P) \wedge \neg\,\text{repl}(P, R) \tag{5.2}$$

In this reaction pair, we call $R$ the replicator and $P$ the parasite. $P$ is parasitic on this particular $R$; it might not be parasitic on other replicators.

## 5.4. Analysis

The logfiles from the runs were parsed with these definitions using the R software package github.com/uoy-research/Rstringmol v. 0.3.1. The program flow of reactions as shown in figure 4 were followed using the Stringmol software and the web app at stringmol.york.ac.uk/webapp.

Data accessibility. The dataset generated and analysed during the current study is available at https://doi.org/10.15124/305dfdb6-9483-4c5b-8a01-c030570b9c31.

Authors' contributions. S.J.H. wrote the code, performed the experiments, did the detailed analyses and prepared the figures. P.H. and S.S. assisted in the interpretation of the results. All contributed to writing the manuscript.

Competing interests. The authors declare that they have no competing interests.

Funding. The Stringmol system was originally developed under the Plazzmid project, EPSRC grant no. EP/F031033/1, and was further developed under the EU FP7 project EvoEvo, grant no. 610427. The simulations were run on the York Advanced Research Computing Cluster (YARCC).

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
