## [Peer Review File · Royal Society Open Science]

Review History

RSOS-210441.R0 (Original submission)

Review form: Reviewer 1 (Wolfgang Banzhaf)

Is the manuscript scientifically sound in its present form?

Yes

Are the interpretations and conclusions justified by the results?

Yes

Is the language acceptable?

Yes

Do you have any ethical concerns with this paper?

No

Have you any concerns about statistical analyses in this paper?

No

Recommendation?

Accept with minor revision (please list in comments)

Comments to the Author(s)

Please see attached file (Appendix A).

Review form: Reviewer 2**Is the manuscript scientifically sound in its present form?**

Yes

Are the interpretations and conclusions justified by the results?

Yes

Is the language acceptable?

Yes

Do you have any ethical concerns with this paper?

No

Have you any concerns about statistical analyses in this paper?

No

Recommendation?

Accept with minor revision (please list in comments)

Comments to the Author(s)

This study by Hickinbotham et al. analyzed an evolutionary simulation using Stringmol system, automata chemistry model which may mimic a replicator in the RNA world, with spatial structure. They found the replicator undergoes an interesting coevolutionary process with parasitic replicators. Initially, replicators became shorter and then developed a parasite-resistant mechanism, but new parasites that circumvent the mechanisms soon appeared. Through the red queen-like dynamics, the replicator became more complex. Because such coevolutionary process may explain the evolution of complexity in the living organisms, I believe this study provides important insight to understand the emergence of life and thus recommend publication after addressing the following comments.

Major points

1. The present title is, I believe, does not appropriately represents the contents of this paper and therefore may be misleading. When I see this title, I thought this is a review article for the general biological phenomena related to evolution with parasites. I highly recommend the authors make a more specific title for this manuscript, which would be helpful for readers to search and grasp the contents of this study easily.
2. In the abstract, the authors wrote "eg by effectively increasing point mutation rates". But I could not find the data on the mutation rates in the Result section.
3. In Fig. 4, some points were difficult to understand. In the "Reactions at t1" panel, the fat arrow on the bottom indicates the reaction from R+P to R+R+P. Why does not P replicate? This point is different from "Reaction at t2" panel, where P replicated (i.e., R+P to R+P+P). Is that a mistake?

The author wrote the gray fat arrow indicates “parasitic reaction”. What is the meaning of “parasitic reaction”? Is that different from the replication of parasites? Please explain more. In this system, all strings must be replicated by another string in my understanding. What is the definition of parasites?

4. In Fig. 5, the usage of “carrying capacity” is confusing. The figure legend shows the blue dotted line is carrying capacity, which is constant for all time points, but the author also wrote in the legend as “Change in reproductive efficiency and carrying capacity for t1-t6). Probably these two carrying capacities are different things. Please clarify.

Minor points

1. In the Results section, two subtitles, “Nothing makes sense...” and “...except in the light of parasites”, are inserted. For me, the insertions are not helpful for reading and unnecessary.
2. P2. Lane 56, right column. “There is little or no corresponding decrease in execution time” is unclear. Is “execution time” is the same as “reaction time” in Fig. 2? If so, I do not still understand the relationship between the average program length and the execution time. Why was the execution time constant with decreasing program length? Please explain more.
3. P3. Lane 44, right column. “Parasites have already emerged at this stage...”. Which color are parasites? In Fig. 3, only lengths are shown in different colors. Please explain how did the authors determine which are parasites from the data.
4. P3. Lane55, right column. “the execution time has hardly changed.” Which data should I see to check this statement? Figure 2? If so, please indicates the time points (t1 to t6) in Fig. 2, which would be helpful for readers to follow the author’s statement.
5. P9. Lane 40, right column. Unnecessary “rate” .

Decision letter (RSOS-210441.R0)

Dear Dr Hickenbotham

On behalf of the Editors, we are pleased to inform you that your Manuscript RSOS-210441 "Nothing in evolution makes sense except in the light of parasites" has been accepted for publication in Royal Society Open Science subject to minor revision in accordance with the referees' reports. Please find the referees' comments along with any feedback from the Editors below my signature.

Please submit your revised manuscript and required files (see below) no later than 7 days from today's (ie 22-Jun-2021) date. Note: the ScholarOne system will 'lock' if submission of the revision

is attempted 7 or more days after the deadline. If you do not think you will be able to meet this deadline please contact the editorial office immediately.

on behalf of Professor Ion Petre (Associate Editor) and Marta Kwiatkowska (Subject Editor)
openscience@royalsociety.org

Associate Editor Comments to Author (Professor Ion Petre):

Associate Editor: 1

Comments to the Author:

Both reviewers point out some concerns they have with the generality of the title of the paper. I urge you to consider their suggestion, but I leave it to you to choose the most suitable title, including the option of keeping it as it is.

Reviewer comments to Author:

Reviewer: 1

Comments to the Author(s)

Please see attached file

Reviewer: 2

Comments to the Author(s)

This study by Hickinbotham et al. analyzed an evolutionary simulation using Stringmol system, automata chemistry model which may mimic a replicator in the RNA world, with spatial structure. They found the replicator undergoes an interesting coevolutionary process with parasitic replicators. Initially, replicators became shorter and then developed a parasite-resistant mechanism, but new parasites that circumvent the mechanisms soon appeared. Through the red queen-like dynamics, the replicator became more complex. Because such coevolutionary process may explain the evolution of complexity in the living organisms, I believe this study provides important insight to understand the emergence of life and thus recommend publication after addressing the following comments.

Major points

1. The present title is, I believe, does not appropriately represents the contents of this paper and therefore may be misleading. When I see this title, I thought this is a review article for the general biological phenomena related to evolution with parasites. I highly recommend the authors make

a more specific title for this manuscript, which would be helpful for readers to search and grasp the contents of this study easily.

2. In the abstract, the authors wrote “eg by effectively increasing point mutation rates”. But I could not find the data on the mutation rates in the Result section.

3. In Fig. 4, some points were difficult to understand.

In the “Reactions at t1” panel, the fat arrow on the bottom indicates the reaction from R+P to R+R+P. Why does not P replicate? This point is different from “Reaction at t2” panel, where P replicated (i.e., R+P to R+P+P). Is that a mistake?

The author wrote the gray fat arrow indicates “parasitic reaction”. What is the meaning of “parasitic reaction”? Is that different from the replication of parasites? Please explain more.

In this system, all strings must be replicated by another string in my understanding. What is the definition of parasites?

4. In Fig. 5, the usage of “carrying capacity” is confusing. The figure legend shows the blue dotted line is carrying capacity, which is constant for all time points, but the author also wrote in the legend as “Change in reproductive efficiency and carrying capacity for t1-t6). Probably these two carrying capacities are different things. Please clarify.

Minor points

1. In the Results section, two subtitles, “Nothing makes sense...” and “...except in the light of parasites”, are inserted. For me, the insertions are not helpful for reading and unnecessary.

2. P2. Lane 56, right column. “There is little or no corresponding decrease in execution time” is unclear. Is “execution time” is the same as “reaction time” in Fig. 2? If so, I do not still understand the relationship between the average program length and the execution time. Why was the execution time constant with decreasing program length? Please explain more.

3. P3. Lane 44, right column. “Parasites have already emerged at this stage...”. Which color are parasites? In Fig. 3, only lengths are shown in different colors. Please explain how did the authors determine which are parasites from the data.

4. P3. Lane55, right column. “the execution time has hardly changed.” Which data should I see to check this statement? Figure 2? If so, please indicates the time points (t1 to t6) in Fig. 2, which would be helpful for readers to follow the author’s statement.

5. P9. Lane 40, right column. Unnecessary “rate”.

===PREPARING YOUR MANUSCRIPT===

Please ensure that you include an acknowledgements' section before your reference list/bibliography. This should acknowledge anyone who assisted with your work, but does not

qualify as an author per the guidelines at <https://royalsociety.org/journals/ethics-policies/openness/>.

===PREPARING YOUR REVISION IN SCHOLARONE===

- Ensure that your data access statement meets the requirements at <https://royalsociety.org/journals/authors/author-guidelines/#data>. You should ensure that you cite the dataset in your reference list. If you have deposited data etc in the Dryad repository, please only include the 'For publication' link at this stage. You should remove the 'For review' link.
- If you are requesting an article processing charge waiver, you must select the relevant waiver option (if requesting a discretionary waiver, the form should have been uploaded at Step 3 'File upload' above).
- If you have uploaded ESM files, please ensure you follow the guidance at <https://royalsociety.org/journals/authors/author-guidelines/#supplementary-material> to include a suitable title and informative caption. An example of appropriate titling and captioning may be found at https://figshare.com/articles/Table_S2_from_Is_there_a_trade-off_between_peak_performance_and_performance_breadth_across_temperatures_for_aerobic_scops_in_teleost_fishes_/3843624.

Author's Response to Decision Letter for (RSOS-210441.R0)

See Appendix B.

Decision letter (RSOS-210441.R1)

Dear Dr Hickinbotham,

I am pleased to inform you that your manuscript entitled "Nothing in evolution makes sense except in the light of parasitism: evolution of complex replication strategies" is now accepted for publication in Royal Society Open Science.

You can expect to receive a proof of your article in the near future. Please contact the editorial office (openscience@royalsociety.org) and the production office (openscience_proofs@royalsociety.org) to let us know if you are likely to be away from e-mail contact -- if you are going to be away, please nominate a co-author (if available) to manage the proofing process, and ensure they are copied into your email to the journal. Due to rapid

publication and an extremely tight schedule, if comments are not received, your paper may experience a delay in publication.

on behalf of Professor Ion Petre (Associate Editor) and Marta Kwiatkowska (Subject Editor)
openscience@royalsociety.org

Appendix A

Comments on “Nothing in evolution makes sense except in the light of parasites”
by Simon Hickinbotham, Susan Stepney and Paulien Hogeweg

This is a very important paper studying the effects of parasitism on evolving replicators in the Stringmol artificial chemistry system. The novelty of this paper is to introduce spatial neighborhood interactions which allows greater diversity and a more realistic model of the evolution of this replicator system. While the model is long established and code is available, this detailed study of the effects of parasitism is a valuable addition to the literature and deepens the understanding of not only this particular model system but the general class of parasite-replicator systems under evolution. The following comments on details of this study are meant to improve the manuscript, but in principle, my recommendation is to publish it, subject to a few minor corrections and additions.

Details

Let me start with the beginning: I don't like the title. I understand that it is taken from a quote by Dobzhansky, and therefore takes authority from there, but is this really necessary? The authors do not provide proof without doubt that such a categorical statement is justified. Then it is not really parallel even to the original. Because Biology is a discipline, evolution a process or mechanism that attempts to explain it, in the original. Here, though, evolution is a process and parasites are objects or entities. You can't explain a process by pointing to objects. So, at least to me, the title does not make sense.

Background Section

Automata chemistries have been examined before, so you might briefly refer to other automata chemistries in the literature, e.g., Dittrich and Banzhaf, “Self-evolution in a constructive binary string system”, *Artificial Life*, 1998. Readers might even benefit from hearing about the bit-string chemistries formulated in the early 1990s by Banzhaf, “Self-replicating sequences of binary numbers”, *Biol Cybernetics*, 1994, which provide an example of the pitfalls of self-replication and the appearance of parasitic interactions. This system was further developed into a 2D spatial system in Banzhaf, Dittrich, Eller, “Self-organization in a system of binary strings with spatial interactions”, *Physica D* 1999. Setting the context of automata chemistries within the larger field of artificial chemistries would be appropriate for this audience. Banzhaf & Yamamoto's book on *Artificial Chemistries* which btw contains a section on Stringmol is relevant in this regard.

Results Section

Fig 2, caption: Is this really averages, or is it the distribution, with median and quartiles being shown?

Fig 3 and discussion of Fig 3:

t1: It would be helpful to mark the point of origin of the replicators you seeded in the 2D landscape.

t2: What is the difference between replicators and parasites - how can you discern them based on length, or other features? The dark blue could be just shorter replicators!

t3: indicted -> indicated

Fig 4, reaction at t6: Where are the bind sites in BBB and Z?

Fig 7: This is a very dense figure, and needs to be well explained. I don't perceive the current explanation as sufficient. Authors should consider adding more explanation in the text.

Discussion of Fig 4, on page 6, ln 56: But the parasite does not toggle, instead, it has a self-scan loop. Where does it copy? It does not seem to read from the replicator again after the toggle.

General comment on results:

What is unclear is how the fact that empty space might not be that widespread after some time and would prohibit proliferation might interact with the various replication/parasite species. I urge the authors to consider studying this question. For example, one other scenario would be that a replication event leads to a pushing out of entities to unoccupied space with the new entity taking their position.

References

References should be augmented by somewhat expanding the discussion in the background section.

Ref 14 is mysterious

Appendix B

Dear sir or madam,

thank you for accepting our paper for publication in *Royal Society Open Science*. Please find below our response to the reviewer's comments, which we have addressed in the final submission

Best regards,

Simon Hickinbotham

Reviewer 1

This is a very important paper studying the effects of parasitism on evolving replicators in the Stringmol artificial chemistry system. The novelty of this paper is to introduce spatial neighborhood interactions which allows greater diversity and a more realistic model of the evolution of this replicator system. While the model is long established and code is available, this detailed study of the effects of parasitism is a valuable addition to the literature and deepens the understanding of not only this particular model system but the general class of parasite-replicator systems under evolution. The following comments on details of this study are meant to improve the manuscript, but in principle, my recommendation is to publish it, subject to a few minor corrections and additions.

Details

Let me start with the beginning: I don't like the title. I understand that it is taken from a quote by Dobzhansky, and therefore takes authority from there, but is this really necessary? The authors do not provide proof without doubt that such a categorical statement is justified. Then it is not really parallel even to the original. Because Biology is a discipline, evolution a process or mechanism that attempts to explain it, in the original. Here, though, evolution is a process and parasites are objects or entities. You can't explain a process by pointing to objects. So, at least to me, the title does not make sense.

Response: The title is intended to emphasise that the results described in the paper imply a major role for parasitism in evolution. We have changed the object 'parasite' to the process 'parasitism', and added an explicit subtitle: **Nothing in evolution makes sense except in the light of parasitism: evolution of complex replication strategies**

Background Section

Automata chemistries have been examined before, so you might briefly refer to other automata chemistries in the literature, e.g., Dittrich and Banzhaf, "Self-evolution in a constructive binary string system", Artificial Life, 1998. Readers might even benefit from hearing about the bit-string chemistries formulated in the early 1990s by Banzhaf, "Self-replicating sequences of binary numbers", Biol Cybernetics, 1994, which provide an example of the pitfalls of self-replication and the appearance of parasitic interactions. This system was further developed into a 2D spatial system in Banzhaf, Dittrich, Eller, "Self-organization in a system of binary strings with spatial interactions", Physica D 1999. Setting the context of automata chemistries within the larger field of artificial chemistries would be appropriate for this audience. Banzhaf & Yamamoto's book on Artificial Chemistries which btw contains a section on Stringmol is relevant in this regard.

Response - thanks, we have added references 10-12 to the introduction.

Results Section

Fig 2, caption: Is this really averages, or is it the distribution, with median and quartiles being shown?

Response we have deleted 'average' from the caption, which now reads:

Change in program length and reaction execution time for molecular interactions. The midlines are the median length/time; the spread shows the inter-quartile range.

Fig 3 and discussion of Fig 3: t1: It would be helpful to mark the point of origin of the replicators you seeded in the 2D landscape.

Response: done, and indicated in the text by adding an opaque square to t1 image of fig 3 to indicate initial distribution of replicators. Description added to figure legend:

The opaque square in the t1 image shows the initial distribution of the hand-coded replicator molecules.

Fig 3 and discussion of Fig 3: t2: What is the difference between replicators and parasites - how can you discern them based on length, or other features? The dark blue could be just shorter replicators!

Response - see response to reviewer 2's minor point 3 below

t3: indicted -> indicated

Response done

Fig 4, reaction at t6: Where are the bind sites in BBB and Z?

Response These have been added to figure 4

Fig 7: This is a very dense figure, and needs to be well explained. I don't perceive the current explanation as sufficient. Authors should consider adding more explanation in the text.

Response we have expanded the explanation by adding the following:

This shows that new parasites emerge as offspring of replicators either directly or as the result of short mutational cascades -- separate lineages of parasitic molecules are not a feature of this system.

Discussion of Fig 4, on page 6, ln 56: But the parasite does not toggle, instead, it has a self-scan loop. Where does it copy? It does not seem to read from the replicator again after the toggle.

Response clarified that 'P' molecule at t3 cannot parasitise the t3 replicator:

the toggle facility to return execution to the replicator and thus access the copy loop. The 'P' molecule in the t3 panel was able to parasitise earlier replicators, but it cannot do this with the dominant replicator at t3, so it is not copied.

General comment on results:

What is unclear is how the fact that empty space might not be that widespread after some time and would prohibit proliferation might interact with the various replication/parasite species. I urge the authors to consider studying this question. For example, one other scenario would be that a replication event leads to a pushing out of entities to unoccupied space with the new entity taking their position.

Response The reviewer makes a good point, but slowing down of replication when an area is full is a reasonable assumption whether it is due to space or food limitation. Pushing away does not work when the arena is really full because there is nowhere to push to. Our conclusion is that this feature is beyond the current implementation but not a significant drawback, and so not worth mentioning in the text.

References

References should be augmented by somewhat expanding the discussion in the background section. **Response** We'll add the Banzhaf-oriented papers suggested by this reviewer

Ref 14 is mysterious **Response** how so? In earlier work we found that the definition of hypercycles is somewhat vague... a related paper we've recently written also raised this issue on review.

- SS: I think they are complaining that the reference doesn't have enough info to track it down, just author/title/year. There is a URL in the BibTeX, but it doesn't show in the ref (note = "`\url{...}`" works better, if you have defined `\url{}`....)

Reviewer 2

This study by Hickenbotham et al. analyzed an evolutionary simulation using Stringmol system, automata chemistry model which may mimic a replicator in the RNA world, with spatial structure. They found the replicator undergoes an interesting coevolutionary process with parasitic replicators. Initially, replicators became shorter and then developed a parasite-resistant mechanism, but new parasites that circumvent the mechanisms soon appeared. Through the red queen-like dynamics, the replicator became more complex. Because such coevolutionary process may explain the evolution of complexity in the living organisms, I believe this study provides important insight to understand the emergence of life and thus recommend publication after addressing the following comments.

Major points

1: *The present title is, I believe, does not appropriately represents the contents of this paper and therefore may be misleading. When I see this title, I thought this is a review article for the general biological phenomena related to evolution with parasites. I highly recommend the authors make a more specific title for this manuscript, which would be helpful for readers to search and grasp the contents of this study easily.*

Response: see response to reviewer 1, above

2: *In the abstract, the authors wrote “eg by effectively increasing point mutation rates”. But I could not find the data on the mutation rates in the Result section.*

Response the ‘self-scan’ phenomena effectively doubles the point mutation rate - text change to emphasise this:

Finally, because self-scan is achieved by using the copy operator (which fires point mutations) to overwrite every opcode with the same opcode, the point mutation rate is approximately doubled for replicating reactions that have this feature.

3: *In Fig. 4, some points were difficult to understand. In the “Reactions at t1” panel, the fat arrow on the bottom indicates the reaction from R+P to R+R+P. Why does not P replicate? This point is different from “Reaction at t2” panel, where P replicated (i.e., R+P to R+P+P). Is that a mistake? **AND** The author wrote the gray fat arrow indicates “parasitic reaction”. What is the meaning of “parasitic reaction”? Is that different from the replication of parasites? Please explain more. In this system, all strings must be replicated by another string in my understanding. What is the definition of parasites?*

Response: Corrections to this figure: Panel t1, corrected the output of the lower reaction to be ‘R+P+P’; added missing bind site graphics to t2 and t6. Caption text for this figure has been edited to clarify that the fat grey arrows indicate non-replicating reactions, not parasitic reactions as had previously been stated:

The fat arrows to the right summarise the reactants and products; the colour represents a replication (red) or other (grey) reaction in which neither of the input reactants are copied; the number shows the program timesteps

4: In Fig. 5, the usage of “carrying capacity” is confusing. The figure legend shows the blue dotted line is carrying capacity, which is constant for all time points, but the author also wrote in the legend as “Change in reproductive efficiency and carrying capacity for t1-t6). Probably these two carrying capacities are different things. Please clarify.

Response We have changed the phrasing from "carrying capacity" to "maximum population size" where appropriate in the caption and discussion of figure 5.

Minor points

1: In the Results section, two subtitles, “Nothing makes sense...” and “...except in the light of parasites”, are inserted. For me, the insertions are not helpful for reading and unnecessary.

Response These subheadings delimit two phases in the analysis: the first phase shows that without considering parasitism the results are difficult to understand; the second shows how parasitism explains the observed dynamics. Admittedly it is perhaps a playful way of doing this...

In line with the change to the title, we have changed “parasites” to “parasitism”

2: P2. Lane 56, right column. “There is little or no corresponding decrease in execution time” is unclear. Is “execution time” is the same as “reaction time” in Fig. 2? If so, I do not still understand the relationship between the average program length and the execution time. Why was the execution time constant with decreasing program length? Please explain more.

Response yes, we had used ‘execution time’ and ‘reaction time’ with identical meanings. We have made sure that the phrase “reaction time” is used throughout. Program length and program execution time are not correlated due to the emergent programmatical features. We have added text to emphasise that short programs can have long execution times:

This explains how short programs can have long reaction times as observed in figure 2 -- a second iteration over the length of the program has been added.

3: P3. Lane 44, right column. “Parasites have already emerged at this stage...”. Which color are parasites? In Fig. 3, only lengths are shown in different colors. Please explain how did the authors determine which are parasites from the data.

Response: The reviewer is correct that program length is not necessarily an indicator of parasitism, hence the need for precise definitions of parasitism. Here we indicate that the labels `replicator' and `parasite' were deduced by further study:

Note that at this point in the analysis, the function of individual molecules in the reactions is inferred by studying the emergent spatial organisation of the system and by reference to dynamics observed in similar studies [23,16,15]. This inference has been confirmed by studying individual reactions as will be described below.

as described later in the text:

In order to determine whether properties like parasitism hold, which require the determination of a counterfactual behaviour (see eqn. 2), we have to perform further analysis of the strings observed at certain epochs. See the subsection "Reaction Types" below for a description of how this is achieved. We summarise the resulting behaviours here.

We have also added a reference to a recent publication [38] in which these issues are discussed in depth.

4: P3. Lane55, right column. "the execution time has hardly changed." Which data should I see to check this statement? Figure 2? If so, please indicates the time points (t1 to t6) in Fig. 2, which would be helpful for readers to follow the author's statement.

Response reference to the appropriate figure has been added (NB this is not figure 2 as suggested by the reviewer, but the top right panel of figure 3)

5: P9. Lane 40, right column. Unnecessary "rate".

Response done.

Finally, please note that the referencing style has been changed to Vancouver